# Plyometric Jump Training Exercise Optimization for Maximizing Human Performance: A Systematic Scoping Review and Identification of Gaps in the Existing Literature

**DOI:** 10.3390/sports11080150

**Published:** 2023-08-09

**Authors:** Ekaitz Dudagoitia Barrio, Rohit K. Thapa, Francisca Villanueva-Flores, Igor Garcia-Atutxa, Asier Santibañez-Gutierrez, Julen Fernández-Landa, Rodrigo Ramirez-Campillo

**Affiliations:** 1Faculty of Sports Sciences, University of Murcia, 30100 Murcia, Spain; ekaitz10@icloud.com; 2Symbiosis School of Sports Sciences, Symbiosis International (Deemed University), Pune 412115, India; rohitthapa04@gmail.com; 3Tecnologico de Monterrey, Escuela Nacional de Medicina y Ciencias de la Salud, Avenida Heroico Colegio Militar 4700, Nombre de Dios, Chihuahua 31300, Mexico; francisca.villanueva@tec.mx; 4Máster en Bioinformática y Bioestadística, Universitat Oberta de Catalunya, Rambla del Poblenou 156, 08018 Barcelona, Spain; igarcia8@uoc.edu; 5Department of Physical Education and Sports, University of the Basque Country, 48940 Bilbao, Spain; asi_santi_10@hotmail.com; 6Health, Physical Activity and Sports Science Laboratory, Department of Physical Activity and Sports, Faculty of Education and Sport, University of Deusto, 48007 Bilbo, Spain; julenfdl@hotmail.com; 7Exercise and Rehabilitation Sciences Institute, School of Physical Therapy, Faculty of Rehabilitation Sciences, Universidad Andres Bello, Santiago 7591538, Chile

**Keywords:** human physical conditioning, exercise, muscle strength, athletic performance, musculoskeletal and neural physiological phenomena

## Abstract

Background: Plyometric jump training (PJT) encompasses a range of different exercises that may offer advantages over other training methods to improve human physical capabilities (HPC). However, no systematic scoping review has analyzed either the role of the type of PJT exercise as an independent prescription variable or the gaps in the literature regarding PJT exercises to maximize HPC. Objective: This systematic scoping review aims to summarize the published scientific literature and its gaps related to HPC adaptations (e.g., jumping) to PJT, focusing on the role of the type of PJT exercise as an independent prescription variable. Methods: Computerized literature searches were conducted in the PubMed, Web of Science, and SCOPUS electronic databases. Design (PICOS) framework: (P) Healthy participants of any age, sex, fitness level, or sports background; (I) Chronic interventions exclusively using any form of PJT exercise type (e.g., vertical, unilateral). Multimodal interventions (e.g., PJT + heavy load resistance training) will be considered only if studies included two experimental groups under the same multimodal intervention, with the only difference between groups being the type of PJT exercise. (C) Comparators include PJT exercises with different modes (e.g., vertical vs. horizontal; vertical vs. horizontal combined with vertical); (O) Considered outcomes (but not limited to): physiological, biomechanical, biochemical, psychological, performance-related outcomes/adaptations, or data on injury risk (from prevention-focused studies); (S) Single- or multi-arm, randomized (parallel, crossover, cluster, other) or non-randomized. Results: Through database searching, 10,546 records were initially identified, and 69 studies (154 study groups) were included in the qualitative synthesis. The DJ (counter, bounce, weighted, and modified) was the most studied type of jump, included in 43 study groups, followed by the CMJ (standard CMJ or modified) in 19 study groups, and the SJ (standard SJ or modified) in 17 study groups. Strength and vertical jump were the most analyzed HPC outcomes in 38 and 54 studies, respectively. The effects of vertical PJT versus horizontal PJT on different HPC were compared in 21 studies. The effects of bounce DJ versus counter DJ (or DJ from different box heights) on different HPC were compared in 26 studies. Conclusions: Although 69 studies analyzed the effects of PJT exercise type on different HPC, several gaps were identified in the literature. Indeed, the potential effect of the PJT exercise type on a considerable number of HPC outcomes (e.g., aerobic capacity, flexibility, asymmetries) are virtually unexplored. Future studies are needed, including greater number of participants, particularly in groups of females, senior athletes, and youths according to maturity. Moreover, long-term (e.g., >12 weeks) PJT interventions are needed

## 1. Introduction

Different resistance training methods have been reported to improve human physical capabilities (HPC) [1,2]. Plyometric jump training (PJT) can offer some advantages over other training methods (e.g., traditional resistance training), offering equal (or even more) effectiveness for the improvement of several HPC (e.g., jumping, sprinting) [3,4]. Indeed, unlike traditional resistance training, the ballistic nature of PJT allows the avoidance of deceleration towards the end of a given movement (e.g., terminal hip and knee extension [5,6]), which might additionally contribute to the transference of adaptations to HPC and sport-specific performance [7,8,9]. Furthermore, PJT may be inexpensive compared to other resistance training methods, requiring little or no equipment, usually involving drills with the body mass used as resistance [10]. Additionally, PJT may be conducted in a relatively small physical space, which may be an essential advantage during specific scenarios (e.g., encountering pandemic restrictions) where participants may be forced to train at their homes [11]. Moreover, PJT may be considered more fun than other training methods (e.g., flexibility, endurance), particularly among youths [12]. Furthermore, PJT may reduce the risk of injury [13,14] and be adapted for successful rehabilitation programs [15]. In addition, PJT can mimic the specific short-duration high-intensity actions of sports, potentially increasing the transference effect between PJT exercises and sport-specific performance [7,8,9]. Indeed, PJT has demonstrated a favorable impact on a myriad of athletes’ physical capabilities, such as jumping, linear running sprinting speed, agility, change of direction speed (CODS), repeated sprint ability (RSA) with and without CODS, short-term endurance (e.g., up to 60 s), long-term endurance (e.g., the Yo-Yo test), maximal strength, balance, sport-specific performance (e.g., kicking speed), range of motion, and coordination, among others [16].

A PJT program compasses a range of exercises that involve high rates of force development and are performed with a variety of ground contact times, ranging from briefer contacts (<250 ms plyometric or fast stretch-shortening cycle [SSC]) [15] as observed during rapid hopping (<200 ms) [17] or hurdle jumps [18] to longer contacts as observed during depth jumping (≥360–400 ms, explosive or slow SSC) [19,20] or the countermovement jump (CMJ; >800 ms) [18]. Indeed, the type of muscle action (e.g., complete SSC [eccentric-amortization-concentric] vs. concentric-only movement; fast vs. slow SSC) may affect the HPC adaptations to PJT. For example, fast SSC PJT drills may exert a more significant effect on a linear sprint (ground contact times [GCT] ~150 ms) and slow SSC PJT drills during actions requiring CODS (GCT ~500 ms during turning movement) [21]. A PJT also involves exercises requiring multi-joint coordination of the lower body and considerable voluntary effort (e.g., near-maximal or maximal vertical jump height) during the concentric portion of a jump against the force of gravity, in addition to the ability to resist strain on the musculoskeletal complex during the eccentric-landing phase [22,23,24]. Indeed, different jumps may involve low (e.g., jump to box) or high (e.g., drop jump) eccentric ground-impact forces that can reach up to 10 times body mass and usually exploit the mechanism of the SSC to augment performance [22,23,24]. Moreover, PJT may involve either unilateral or bilateral leg movements, without external load (e.g., body mass load) or with external load (e.g., loaded CMJ, jump squat), with different potentials to affect the force–velocity profile [25]. A PJT program also involves exercises with varying directions of force application (e.g., vertical vs. horizontal), which may affect the degree of HPC adaptation. For example, vertical-predominant jump training may significantly impact HPC with a greater vertical component (e.g., vertical jump). In comparison, horizontal-predominant PJT may have a greater effect on HPC with a greater horizontal component (e.g., linear sprint) [26]. Furthermore, the specificity of the PJT exercise concerning the targeted outcome, and the inter-repetition pattern (e.g., cyclic vs. acyclic) [27], may additionally affect HPC adaptations.

Because of these variations, a wide array of PJT exercises are available to physical conditioning coaches to facilitate a range of HPC adaptations in line with manipulating parameters such as training intensity, frequency, or jump repetitions [28,29,30]. Although there is a reasonable amount of scientific literature on the effects of the type of PJT exercise on HPC adaptations, considering the myriad of PJT exercise variations that are possible [31,32,33], it is likely that a majority of the PJT types that could be incorporated into a training program have not been adequately investigated. Indeed, coaches’ decisions regarding potentially relevant PJT moderators are frequently based on practical experience or evidence from cross-sectional studies with particular populations [34]. Moreover, experimental research approaches in PJT studies usually compare a limited number of PJT exercises. Indeed, PJT studies commonly include only two or three groups of participants, and a control group is not always available. Furthermore, most PJT studies involved only small samples of participants (i.e., *n* = 10) [3,4], precluding a generalization of results to broader groups [35]. In this context, an alternative research approach to better analyze the effect of a broader range of PJT exercises may involve a systematic literature review.

Systematic reviews may assist practitioners in selecting more effective and safer PJT prescriptions through an evidence-based decision approach [36,37,38]. Although some systematic reviews with meta-analyses attempted to analyze the role of potentially relevant PJT moderators (e.g., PJT intensity) on HPC [39,40], analyses on PJT exercise type were usually precluded due to an insufficient number of studies available. Relatedly, systematic reviews, with and without meta-analyses, involve inherent strict inclusion criteria [41,42,43], usually restricted to randomized-controlled studies. However, such a research design can be logistically challenging in PJT studies, particularly with highly trained athletes. This would involve the exclusion of such studies from systematic reviews. Thus, much of the evidence in this field would be limited to analyses, precluding a more comprehensive analysis regarding the potential effects of PJT on HPC. An alternative approach to a traditional systematic review would involve a systematic scoping review.

Scoping reviews perform a systematic mapping of existing evidence and identify relevant gaps in the literature [44,45]. Scoping studies aim to provide more than pooled results or analytical comparisons by also mapping the existing evidence [45]. Future research would benefit from clear guidance based on an evidence-gap map (EGM) [46,47], and scoping reviews provide a suitable and systematic approach to building such maps [45]. Fitting into the broad approach of most scoping studies, EGMs graphically represent the body of evidence, conveying an intuitive visual interpretation of research efforts allocation (i.e., where the evidence is rich versus where it is scarce) [46,47,48]. Such data assists in developing policies and guidelines and exposes areas requiring further research [46,47,48]. Sports-medicine-related reviews, including EGMs, have been published in recent years [49,50,51]. A scoping review with an EGM will provide a clearer picture of what is known about PJT exercise type, as a prescription variable, for physical performance maximization in healthy participants, helping inform future policies and funding.

Previous systematic scoping reviews [3,4,52] have addressed PJT programming issues. However, these studies included a broad scope, not focusing on the potential role of the type of PJT exercise on HPC, concentrating on a particular group of participants (e.g., soccer players). Additionally, the rate of yearly PJT-related publications increased 25-fold between 2000 and 2017 [3]. More frequent updates are necessary for sports science. Moreover, the increasing number of publications in PJT will likely render prescription reviews quickly outdated. In rapidly emerging research fields, 25% of systematic reviews are obsolete within two years and 50% within five years. Periodic systematic review updates of the literature (a systematic living review of the literature) have been recommended to cope with fast-growing fields of knowledge [53]. The main advantage of this approach is that it assumes that new knowledge will improve sports and clinical decision making [53]. As such, a continuous systematic review update based on the new relevant evidence seems a good option [54,55]. Such a potentially suitable method has yet to be applied in the field of PJT effects on HPC and the potential moderator role of the PJT exercise type.

Considering this rationale, this article aims to summarize the published scientific literature related to HPC adaptations (e.g., jumping) to PJT, focusing on the role of the type of PJT exercise as an independent prescription variable, using a systematic scoping review approach. Therefore, this systematic scoping review would add to the literature by grouping the vast number of studies, independent of the study design (i.e., controlled, not controlled, randomized), involving PJT interventions to improve HPC performance. Although previous scoping reviews have addressed the role of PJT, none have included a particular focus on the role of the type of PJT exercise as an independent prescription variable on a broad number of HPC and groups of participants. This review approach would add valuable information to the literature for practitioners and applied researchers.

## 2. Methods

### 2.1. Procedures

A systematic scoping review was conducted following previous guidelines, including the Preferred Reporting Items for Systematic Reviews and Meta-Analyses (PRISMA 2020) and PRISMA extension for Scoping Reviews [44,56,57,58,59].

### 2.2. Literature Search: Administration and Update

We considered recommendations from systematic scoping reviews that previously examined the PJT literature [3,4]. Computerized literature searches were conducted in PubMed, Web of Science, and SCOPUS electronic databases. The search strategy was performed using the Boolean operators AND in different combinations with keywords for all database fields (i.e., “ballistic”, “complex”, “cycle”, “force”, “plyometric”, “shortening” “stretch”, “training”, “velocity”) or title database field (i.e., “explosive”, “jump”, “power”, “training”). These were combined as (i) “ballistic” AND “training”, (ii) “complex” AND “explosive” AND “training”, (iii) “explosive” AND “training”, (iv) “force-velocity” AND “training”, (v) “jump” AND “training”, (vi) “plyometric” AND “training”, (vii) “power” AND “training”, and (viii) “stretch” AND “shortening” AND “cycle” AND “training”. After an initial investigation in April 2017, an account was created by one of the authors (RRC) in each of the respective databases, through which the author received automatically generated email updates regarding the search terms used. The search was refined in May 2019 and August 2021, with updates received daily (if available). Studies were eligible for inclusion up to October 2022. The main advantage of this search approach is that it assumes that new knowledge will appear and allow improvements in sports/clinical decision making [53,54,55]. Indeed, the rate of PJT studies published yearly has increased exponentially since 2010 [3,4]. The same author (RRC) conducted the initial search and removed duplicates using the automated removal function of duplicates of EndNote^TM^ 20.4.1 for Windows (Clarivate^TM^), with further manual removal of duplicates if required. After that, the search results were analyzed according to the eligibility criteria. The electronic Appendix A describes the search strategy (code line) for each database and the background of the search history (Appendix A).

In selecting studies for inclusion, all relevant titles were reviewed before examining the abstracts and full texts. Then, a double screening was performed [60]. First, one experienced researcher (RRC) independently screened the retrieved studies’ titles, abstracts, and full texts, with a second author (ED) confirming. Potential discrepancies between the two authors regarding inclusion and exclusion criteria (e.g., intervention adequacy) were resolved through consensus with a third author (RKT) during the search and review process). After that, the list of included studies and the inclusion criteria were sent to two independent world experts in the field of PJT, identified through the “Plyometric Exercise” field in Expertscape^®^. Due to a large number of expected studies, there may have been reduced compliance from the experts, especially since (by definition) they cannot be invited to be coauthors of the manuscript (otherwise, they would not be independent experts). A three-week waiting period was granted for the 1st response (including a reminder after the first two weeks) and an additional four-week period for completing the task in case of a positive response. Upon having the final list of included studies, we manually searched for errata and retractions [61] and retrieved pre-registered or pre-published protocols and supplementary files when available. Snowballing citation tracking was not performed due to the large number of studies expected to be included in this systematic scoping review. If the number of initially included studies had proved to be not enough to provide representative data on past and current trends in this field (i.e., <100 studies), with further studies likely making an impact on our results, manual searches would have been performed within the reference lists of the studies deemed eligible for inclusion after the automated searches. We also selected representative systematic reviews on the topic and checked their reference list.

### 2.3. Inclusion and Exclusion Criteria

Research articles published in peer-reviewed journals were considered, with no publication date or language limitations. Eligibility criteria were based on the Participants, Intervention, Comparators, Outcomes, and Study Design (PICOS) framework [56]: (P) Healthy participants of any age, sex, or sport. Studies with injured (e.g., studies on rehabilitation or return to sports) were excluded; (I) Chronic (multiple sessions with an assessment of pre- to post-differences) interventions exclusively using any form of PJT exercise type (e.g., vertical, unilateral), either single mode (e.g., vertical only) or combined mode (e.g., vertical combined with horizontal PJT exercises). Multimodal interventions (e.g., PJT + heavy load resistance training) were considered only if studies included two experimental groups under the same multimodal intervention, with the only difference between groups being the type of PJT exercise. An evidence-based [3,4] decision was considered to determine the minimal effective PJT duration (weeks) for the improvement of HPC, i.e., ≥2 weeks; (C) Comparators include PJT exercises with different modes (e.g., vertical vs. horizontal; vertical vs. horizontal combined with vertical); (O) Considered outcomes (but not limited to): physiological, biomechanical, biochemical, psychological, performance-related outcomes/adaptations, or data on injury risk (from prevention-focused studies); (S) Single- or multi-arm, randomized (parallel, crossover, cluster, other), or non-randomized.

Only original studies in peer-reviewed and full-text format were eligible to be included. Additional exclusion criteria: books, book chapters, and congress abstracts, as well as cross-sectional and review papers, and training-related studies that did not focus on the effects of PJT exercises, such as plyometric training without the use of jumps (e.g., upper-body plyometrics only). Also excluded were retrospective studies, prospective studies (e.g., the relationship between bone density at the end of PJT and several years of follow-up), studies in which the use of PJT exercises was not clearly described (e.g., authors stated “plyometric exercises were implemented”, without further explanation), studies for which only the abstract was available, case reports, special communications, letters to the editor, invited commentaries, errata, studies with questionable quality or unclear peer-review process from the journal [62], overtraining studies, and detraining studies. In the case of detraining studies, these were considered for inclusion if they involved training before a detraining period. Because of expected difficulties with the translation of research articles written in different languages and the fact that 99.6% of the PJT literature is published in English [3], only articles written in English, Spanish, German, and Portuguese (i.e., authors’ native languages), were considered for inclusion.

## 3. Data Extraction

### 3.1. Data Collection Process

Being a systematic scoping review, data refer to study characteristics and their outcomes but do not include the actual data results derived from specific tests-measurements, which were not extracted. All data was coded into a specifically designed Microsoft^®^ Excel worksheet. If relevant data or contextual information proved to be missing, the studies’ authors were contacted through email, and a three-week waiting period was granted for the response (including a reminder after the first two weeks). Without a response within three weeks, the study was excluded if the needed information was required according to eligibility criteria. If the missing information was not integral to the eligibility criteria, the study was included in the review.

### 3.2. Data Items

Participant-related information: sample size, age, sex, sport, season timing (e.g., pre-season, competitive phase), fitness level, body mass, height, and previous experience with PJT.

Intervention-related information focused on chronic adaptations: intervention length, PJT exercise type (e.g., vertical), identifying the box height when appropriate (e.g., PJT involving drop jump exercise); repetitions; intensity; frequency; co-interventions (e.g., PJT combined with heavy resistance training); inter-repetition, inter-set, and inter-day recovery time; type of surface; progressive load dose; application strategy (e.g., replaced a portion of formal training with PJT); and tapering strategies.

Comparators: other PJT exercise types, i.e., in the same study, two groups should be included in the PJT intervention, with the only difference between the groups being the type of PJT exercise used during the intervention period.

Outcomes: physiological (e.g., muscle fiber diameter), psychological (e.g., rate of perceived exertion [RPE]), HPC (e.g., CMJ height; CMJ force; a range of motion), and system level(s) (e.g., cardiovascular, musculoskeletal, nervous). The HPC outcomes will also be analyzed according to their factor emphasis (e.g., strength, flexibility) to provide an overview of which categories are being assessed. Considering the goal of providing a systematic scoping review with an evidence-gap map, outcomes were registered, but their results were not. For example, k studies assessed the CMJ, but the actual measurement values were irrelevant.

Study design-based evidence-level: a color coding denoted randomized (green) and non-randomized multi-arm (yellow) studies. Considering the purposes of this systematic scoping review, analyzing the risk of bias in studies would not impact our results or the assessment of research trends [63].

One author (RRC) performed data extraction, and a second author (ED) provided confirmation, and any discrepancies between them were resolved through consensus with a third author (RKT).

### 3.3. Data Management and Synthesis Methods

A narrative synthesis was performed, accompanied by data summaries (e.g., number, percentage) for the previously defined data items to provide an overview of the existing body and the corresponding gaps in research. An EGM was constructed to graphically represent the body of evidence and intuitively convey an overview of the existing evidence and the current research gaps [46,47,48].

### 3.4. Registration and Protocol

The protocol was pre-registered in Open Science Framework (OSF). The first reference given by OSF was: Barrio, E. D., Thapa, R. K., & Ramirez-Campillo, R. (20 October 2022). What don’t we know about plyometric jump training exercise type optimization, as a prescription variable, for human performance maximization: A systematic scoping review with evidence-gap map. “https://doi.org/10.17605/OSF.IO/Q2Y3A (accessed on 4 August 2023)”.

## 4. Results

Figure 1 provides a graphical schematization of the study selection process. Through database searching, 12,503 records were initially identified, and 69 studies were included in this systematic scoping review. The Appendix A presents the studies excluded (with reasons) at the preliminary qualitative synthesis stage.

The 69 studies included appeared in 37 different journals, all written in English, with an exponential increase in the number of published articles per year in recent years. Table 1 summarizes the articles included in this systematic scoping review. Figure 2 shows the number of articles included accumulated (grouped) over periods of five years (Figure 2).

### 4.1. Participants’ Characteristics and General Critical Elements of Plyometric Jump Training

Table 2 shows the participant characteristics from the 69 studies included. The range of participants’ age was 11 to 39 years, with a mean of 20.1 years. Participants’ mean body mass, stature, and body mass index were 69.5 kg, 174.1 cm, and 22.9 kg.m^−2^, respectively. The rest of the relevant information is included in Table 2.

Table 3 shows the general critical elements of PJT. The underfoot surface type was not reported in 73.9% of studies (51 of 69). Regarding soft surfaces, 10.1% (7 of 69) of studies reported the use of grass, 4.3% used athletic mats (3 of 69), 2.9% (2 of 69) used sand, and only 1.4% (1 of 69) reported unstable surfaces. Three studies (2.9%) used special equipment (e.g., force plates or different machines) to perform PJT, and only one (1.4%) used a mixture of both types of surface (mat vs. wooden parquet). Concerning the total dose of interventions (e.g., foot contacts per leg, number of jumps, time, velocity, strength, etc.), 97.1% of studies report this information. A wide range of values was observed, from 137 to 3888 jumps. However, values varied according to training design (e.g., duration). PJT was combined with other training methods as part of an intervention in 17.4% (12 of 69) of studies, but no clear information was identified in 7.2% (5 of 69) of the studies. In most studies, combined resistance training was used the most in 66.7% (8 of 12) cases. Volleyball, physical education classes, and combined sprint, resistance training, and feedback were the other methods combined with PJT [33.3% (4 of 12)]. However, in most included studies [75.4% (52 of 69)], the PJT intervention programs were not combined with any other type of training. Training duration ranged from 3 to 12 weeks. A total of 79.7% of studies applied weeks of training (mode, in 21 of 69), with a mean of 7.1 weeks observed. Regarding training frequency, this ranged from 1 to 5 days per week; 55.1% of studies used 2 days per week, and 30.4% used 3 days per week. Only 8.6% applied a combination of training frequencies, commonly two and three sessions per week. PJT intensity was not clearly reported in 26.1% of the studies included. In comparison, 60.9% reported it as maximal using criteria such as height, distance, reactive strength index, optimal power, percentage of one repetition maximum, time, voluntary effort, velocity, rate of execution, force, or a mixture of these. Only 13.0% used submaximal intensity, quantified as the percentage of one repetition maximum, height, distance, velocity, and rating of perceived exertion. The rest time between sets and/or exercises was not clearly reported for 21.7% of the studies. The rest interval extended from 30 to 600 s, with a mean of 132 s and a mode of 120 s (14 of 69). With regard to the rest period between plyometric jump repetitions, 66.7% of the studies did not specify the interval or were not applicable. For those that reported the duration, this ranged from 2 to 30 s, with a mean of 11 s and a mode of 15 s (8 of 69). The rest period between training sessions was not reported in 50.7% of the studies. Among those studies that reported this value, 48 and 72 h were the most typical rest period durations reported, with intervals ranging from 24 to 120 h.

### 4.2. The Type of PJT Exercise as an Independent Prescription Variable

All of the 69 included studies recruited two or more intervention groups, for a total of 154 study groups, and 33.7% of groups mixed different jumps during the intervention. Thus, 66.2% employed a kind of jump only [mostly CMJ (13.7%) or DJ (30.0%)]. Box heights for DJs ranged from 10 to 110 cm, and individualized prescription of heights was used in 6.2% (10 groups). The type of PJT exercise prescription was grouped into 33 different groups to show this analysis. Figure 3 includes an EGM of the 154 study groups grouped by type of jump employed and study design (e.g., randomized-controlled, randomized non-controlled, non-randomized controlled, and non-randomized non-controlled) (Figure 3). DJ (counter, bounce, weighted, and modified) was the most studied type of jump included in 43 groups, followed by CMJ (usual CMJ or modified) in 19 groups and SJ (usual SJ or modified) in 17 groups.

### 4.3. Comparisons of Plyometric Jump Training Exercises on Selected Outcomes of Human Physical Capabilities

Table 4 shows an EGM of PJT exercise type and outcomes measured in terms of HPC. Vertical jump and strength HPC outcomes were the most analyzed in 54 and 38 studies, respectively. Sprint, power, agility, physiological measurements, and horizontal jump performance were followed by 22, 22, 18, 16, and 12 studies, respectively. The least measured results related to HPC were biomechanical-related, sport-specific performance, balance, aerobic, asymmetry, and flexibility, by 7, 6, 3, 2, and 2 studies, respectively.

Bounce versus counter DJ or DJ using different box heights was the most used, 26 times, followed by vertical versus horizontal jumps comparison, 21 times. Bilateral versus unilateral and DJ versus CMJ comparisons were studied 21 times each. Assisted versus resisted and fast SSC versus slow SSC jumps were compared 15 and 11 times, respectively. The rest of the comparisons were measured fewer than ten times and were included in the EGM (Table 4).

The DJ results seem similar to those of CMJ in terms of improving vertical jump height; six studies compared these types of jump, with four noting similar improvements with both training prescriptions [81,105,106,122], one favoring DJ [77] and another CMJ [110].

No differences were found in any studies comparing bounce vs. counter DJ to improve lower limb strength [68,79,87,98,116,129]; only one favored the group that used optimal RSI box height [107]. To improve vertical jump performance, two studies favored counter DJ [95,116]; however, the other seven did not show differences between these comparators [70,79,87,98,107,121,129].

Regarding unilateral vs. bilateral jump comparators on vertical jump performance, unilateral jumps are superior to bilateral jumps in three of five studies that evaluated this [69,91,103]; in the other two, the group that mixed both types was better in one [108], and in the other, there were differences between groups [118].

Vertical jump performance was similar in vertical vs. horizontal jump in three studies [26,83,92], while one favored the vertical jump group [114]. Four studies measured sprint performance; one favored the horizontal training group [124], another a group that mixed both types of exercises [26], and the final two found no differences [92,113].

## 5. Discussion

This scoping review with EGM aimed to summarize the latest scientific literature related to HPC adaptations (e.g., jumping) to PJT, focusing on the role of the type of PJT exercise as an independent prescription variable, using a systematic scoping review approach. The main results comprehensively characterize the leading HPC regarding PJT exercises. The following paragraphs discuss the identified gaps and future directions for the PJT type of exercise research regarding HPC.

### 5.1. General Characteristics

From the 69 eligible articles that included a minimum of two experimental groups to perform different types of PJT, 38.5% needed to be more clearly described, meaning that their findings could not be leveraged for putting into practice or being reproduced by scientists with better methodologies. An insufficiently described study implies the omission of treatment descriptors, such as training duration, frequency, intensity, etc. Thus, 61.5% of the studies included in this review demonstrated a high description quality, and their findings are the line to follow in future research. However, in terms of the results, only 50.7% of the studies included reported at least one dependent variable mean change between pre- and post-intervention. Also, given the growing consensus concerning the importance of effect sizes in intervention studies, it is relevant to include this measure [130]. Regrettably, only 23.2% of studies reported this measure clearly, and these values were often presented in graphical form or not registered. Although most of the included studies had well-described methodologies, investigators should try to isolate as many conditions as possible for performing the types of jumps described. For example, a study to compare horizontal vs. vertical jumps in basketball players is usually contaminated by the sport practiced, basketball, which involves many vertical jumps in both training groups. Thus, including an active control group and another passive control group could be interesting. A crossover design could be an optimal alternative when using a control group is not possible due to a small sample size or other reasons. However, this may be a suboptimal approach for athletes physically maturing fast [131].

### 5.2. Characteristics of Participants and General Critical Elements of Plyometric Jump Training

Another scoping review that involved all PJT studies identified as a shortcoming the poor number of studies conducted with females (only 22%) [4]. Only 24.6% included <18 years old samples; this indicated a gap in the literature in studies that compare different types of jumps in youth participants. This gap has been previously reported among PJT and resistance training studies [3,132]. It is known that biological age influences adaptations to PJT interventions [133]. However, only a few consider the maturation phase using tools like the Tanner scale. At the same time, the oldest group reported was 39 years old; this indicates a big literature gap comparing jump types on older adults. Strength and conditioning professionals need to know much more about the jump exercise selection for a good training prescription for older adults in terms of HPC. Regarding anthropometric measures, all subjects were healthy and within usual standards; no studies were carried out with overweight participants. Thus, despite PJT improving motor performance in obese young boys and metabolic abnormalities in obese females [134,135], the current literature does not present data about the type of jumps in this population.

Similar to the scoping review results that include all PJT studies [3], only 14.5% of the articles included were conducted with high-level participants. Although PJT seems effective in improving athlete performance [136], the lack of a high-level sample could be due to professional trainers’ refusal to modify training sessions or transfer data to others. In addition, previous experience with PJT in the sample (only 20.3% of studies included) could impact the training adaptation and deliver lower benefits due to the high requirements of training that need experienced athletes. Subjects without previous experience improve their performance more quickly due to the new stimuli demanded, which could be better for obtaining significant results. Most participants involved in a competitive season participated in studies during the in-season or pre-season (30.4%); only 2.3% were carried out during the off-season, despite the benefits of implementing PJT during this period for eliciting strength and power gains [80]. That could be due to the difficulty of recruiting and monitoring enough participants during this period. However, although in-season implementation of PJT could interfere more with regular training, its application could reduce the risk of injuries, especially among youth athletes [137]. Strength and conditioning coaches can monitor their athletes more precisely during in-season and pre-season periods than during the off-season, which could explain the difference in these results.

To show the specific effect of one type of jump over another, isolating one unique exercise is the better choice. A total of 66.25% of the groups included in this review performed one unique exercise per training group; this is an excellent scientific strategy to observe the effect of one type of jump vs. another. Nevertheless, on rare occasions, athletes or casual physical activity users use only one exercise in their exercise programs. Thus, for strength and conditioning professionals, studies and training groups that perform more than one different jump of the same type could be more helpful for their work. For example, a study that compares various vertical jumps vs. various horizontal jumps could be more representative of the training programs than one that only compares one vertical jump vs. one horizontal jump [92]. Another essential tip to better assess different types of jumps is to report if the study training methodology was added to or replaced the regular training of the participants [138,139]. However, 36% of studies do not report this, and 35% added the study training methodology to participants’ regular training, which could make the improvements ascribed to the PJT more questionable. Relatedly, 17.4% of the included studies reported that the PJT training was added to another training method as part of an intervention (resistance training, volleyball, physical education classes, or sprinting). Therefore, researchers should consider these methodological limitations to draw accurate conclusions. Regarding HPC, asymmetries, flexibility, and aerobic capacity were the most significant gaps regarding PJT exercise type, with only two comparators for each of these HPC, followed by balance with three comparators.

### 5.3. The Type of PJT Exercise as an Independent Prescription Variable

The type of PJT exercise prescription was grouped into 33 different groups to show this analysis. This aggregation was created to show researchers the most studied PJT exercises and the primary characteristics of their studies. For example, the modified CMJ group included CMJs that refrain from moving specific joints (e.g., no arm movements, no knee flexing, etc.) or were performed de-loaded or loaded. In this sense, counter DJ was the most analyzed exercise included in 25 groups. The literature shows a robust analysis of this exercise, with 16 of 25 groups analyzed presented randomized controlled trials. However, from a practical perspective, to optimize HPC, isolating only one type of jump is not the best choice [26]. Figure 3 shows a quick view of the literature PJT exercises analyzed and the robustness of the evidence. The longer the column, the more researched the type of jump is, and the darker the column, the higher the quality of the evidence. One group was found in the literature that uniquely used a mix of horizontal, unilateral exercises or a mix of vertical, bilateral. In addition, researchers could consider other types of exercises that still need to be explored, e.g., acyclical unilateral or loaded unilateral exercises.

### 5.4. Comparisons of Plyometric Jump Training Exercises on Selected Outcomes of Human Physical Capabilities

Table 4 shows an EGM comparing PJT exercise types and outcomes measured in terms of HPC. Blank squares represent comparisons that have yet to be studied. For example, CMJ vs. arms-restricted CMJ; the literature does not show us anything about strength, horizontal jump, sprint, COD, power, asymmetries, SSP, physiological changes, flexibility, balance, and aerobic capacity HPC data in this comparison of exercise type.

Literature studies that showed more than four comparators on the same HPC were analyzed for vertical jump performance, comparing DJ vs. CMJ; in four studies, no differences in performance between the type of jumps were found. In these four studies, the measure used to assess vertical jump performance was CMJ; however, in two, the group trained with CMJ exercised in the sand [105,106]. It is important to consider the surface type because it is a determinant that induces specific adaptations [139]. The study which favored the CMJ group involved female volleyball players and measured specific jumps in volleyball. The authors reported that the advantage of this type of jump was due to slower SSC characteristics and seemed more sport-specific [110]. However, in a study that involved subjects not involved in competitive sports or recreational activities involving jumps, the DJ seemed more effective than CMJ in improving vertical jump in DJ, CMJ, and CMJ and DJ [77]. Thus, subjects that usually were not involved in fast SSC activities could be more sensitive to this type of stimulus than to slow SSC activities. Regarding bounce and counter DJ, seven studies did not show differences in favor of the jump-measuring of DJ or CMJ height [70,79,87,98,107,121,129]. However, one study included a group with no fixed DJ height and individualized each subject to their maximum RSI; these groups performed better than groups with fixed box height [107]. Another study, which included a box height a bit higher than optimal box height but performed these jumps with loads, showed similar improvements in the group with optimal box height and better vertical jump performance to those using less than optimal box height [116]. In contrast, a study carried out by Marshall and Moran et al. [95] compared purely bounce vs. counter DJ jumps with the same height of box but changing the instructions to participants (e.g., jumping more quickly vs. jumps at maximum height) and they discovered that counter DJ was more effective than bounce DJ at enhancing CMJ height. Unilateral jumps seem to be more sensible for improving vertical jump performance than bilateral jumps [69,91,103]. However, a study that mixed both types of jump showed better performance, so a combination seems more advantageous [108]. The study carried out by Makaruk et al. [91] suggested that unilateral exercises produce better jumping performance in a shorter period compared to bilateral exercises. However, achieved performance gains last longer after bilateral PJT. So, these conclusions could be used by strength and conditioning coaches depending on their goals and the need for short or extended periods. The orientation of jumps, vertical vs. horizontal, is indifferent in improving CMJ height [26,83,92], and a combination of the two seems to be advantageous [26]. In the only study that assessed DJ height as a vertical jump parameter, vertical jumps proved better than horizontal jumps [114]. So, in addition to specifying the direction of jumps, the characteristics of fast or slow SSC could be better specified to induce adaptations.

To assess the strength outcome, the only comparator with enough studies was bounce vs. counter DJ. However, no differences were shown in tests involving 1RM leg press, 1RM knee extension, MVC, etc., [68,79,87,98,116,129]. The exception was a study that included a group that used optimal RSI box height vs. fixed, which proved better for improving 5RM squat [107]. So, again individualization could be key to prescribing PJT. To assess sprint outcome, the only comparator with enough studies was vertical vs. horizontal. Once again, only the study that mixed a group with both types of jumps showed advantages [26], so individualized and mixed different kinds of hops with different characteristics may suppose better stimuli.

Aerobic capacity, flexibility, and asymmetries are the outcomes that have received minimal attention from researchers, with only two studies conducted for each. Similarly, balance has been a considered outcome in only three studies. However, the dynamic nature of plyometrics requires increased oxygen uptake and energy utilization, which may contribute to some extent to improving aerobic capacity [140]. Plyometric exercises often involve stretching and lengthening muscles before the explosive contraction phase. The repeated stretching and loading of muscles during plyometric movements could improve flexibility [141]. Plyometric training challenges the neuromuscular system and requires athletes to control their body movements in various planes of motion. This constant demand for stability and coordination during plyometric exercises can improve balance and proprioception (awareness of body position in space). Plyometric exercises often involve bilateral and unilateral movements, which can help address asymmetries by promoting equal strength and coordination on both sides of the body.

## 6. Limitations

Despite the comprehensive nature of this systematic scoping review, which encompassed numerous articles comparing various PJT exercises, it is essential to acknowledge certain inherent limitations. The limited data analysis: this review did not conduct statistical analyses or meta-analyses on the results of individual articles. Consequently, the assessment of the performance and health impacts associated with each type of jump will be addressed in future research endeavors. The lack of a specific research question: this scoping review adopted a broad approach and did not center on specific research questions. However, these aims were successfully achieved given that the objective was to identify gaps in the literature, highlight areas for future research, and provide a comprehensive overview of PJT exercises.

## 7. Conclusions

Exploring the literature gaps on HPC adaptations through PJT exercises reveals the need for comprehensive, high-quality research across various domains (see Table 2 and Figure 3). Notably, the vertical jump is the most extensively investigated aspect of HPC, with an impressive 54 comparative studies, followed by strength with 38 studies. Conversely, outcomes in terms of aerobic capacity, flexibility, and asymmetries have received minimal attention from researchers, with only two studies conducted for each. Similarly, balance was a considered outcome in only three studies. Notably, a handful of PJT exercise comparisons have received considerable attention, with four or more studies conducted. These include DJ vs. CMJ (focused on strength), bounce DJ vs. counter DJ (focused on strength and vertical jump), bilateral jumps vs. vertical jumps (focused on vertical jump), and vertical jumps vs. horizontal jumps (assessing vertical jump and sprint performance). Yet, the breadth of unexplored territory in this field remains substantial, urging researchers to illuminate and deepen our understanding of PJT exercises in the context of HPC. As the authors of this systematic scoping review, we offer this work as a guiding resource for future investigations in sports sciences, intended to bridge the identified literature gaps. Moreover, researchers will find it invaluable in determining gaps in PJT exercise selection, providing a roadmap for future innovative research endeavors.

## Figures and Tables

**Figure 1 sports-11-00150-f001:**
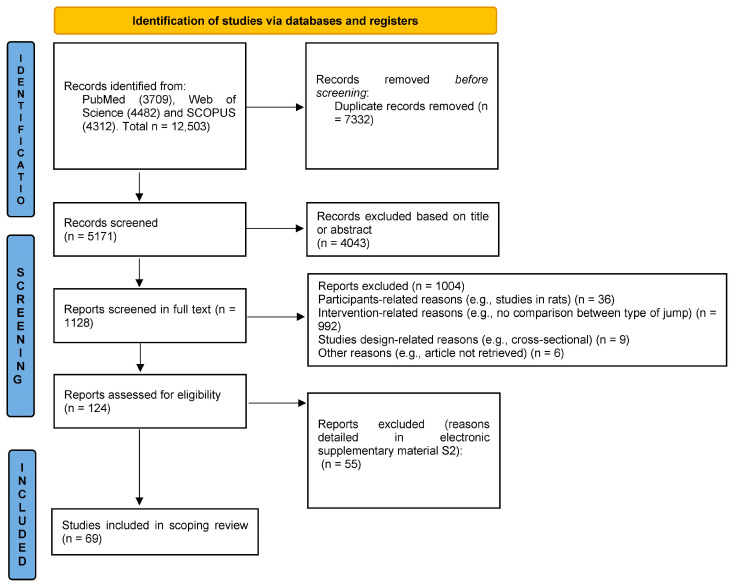
PRISMA flow diagram of the identification of studies.

**Figure 2 sports-11-00150-f002:**
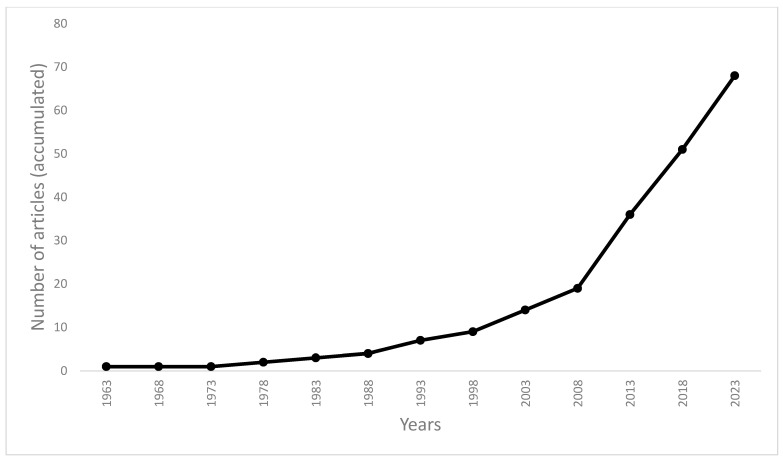
Number of included articles (accumulated) as per year of publication.

**Figure 3 sports-11-00150-f003:**
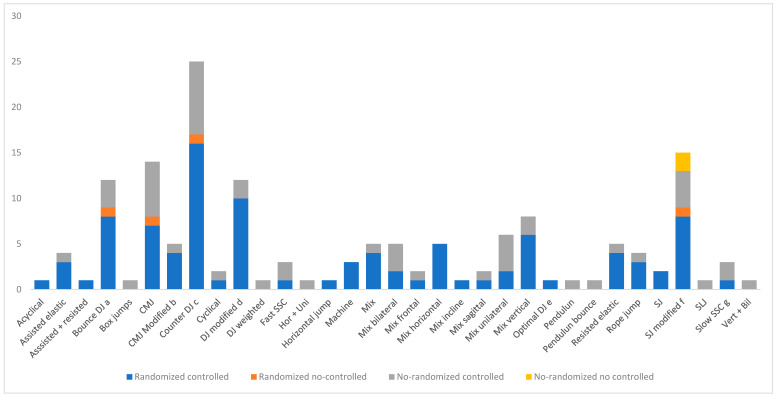
PJT exercise as an independent prescription variable. Bounce DJ a: focus on reduced foot–ground time contact; CMJ Modified b: refrain from moving specific joints (e.g., no arm movements, no knee flexing, etc.) or perform de-loaded or loaded; Counter DJ c: focus on maximizing jump height. (studies in which instructions were to minimize ground contact time and maximize height were included as Counter DJ); DJ modified d: focus on specific joints (e.g., ankle, knee, or hip) or in the following jump (e.g., horizontal or vertical); Optimal DJ e: height that elicited the highest ratio of jump height to contact time; SJ modified f: reduce the joint range of movement or loaded SJ (e.g., only concentric loaded, only eccentric loaded, or both); Slow SSC g: slow or suprime SSC.

**Table 1 sports-11-00150-t001:** Description of studies included.

Study	Randomization	Sample Size	Gender	Age	Freq	Dur	Box Height	Total Jumps	Type of Jump Training	Combined	Tests
Abass (2009) [64]	Yes	10	Male	24.9	3	12	35, 40, 45	NR	Depth jump	No	- Back and leg maximal strength
		10		24.9			NA		Rebound jump		
		10		27.5			NA		Horizontal unilateral		
Andrew et al. (2010) [65]	Yes	12	Mix	22.3	2	12	15–60	2016	Hip depth jump	No	- One- and two-legged vertical jump, 30 m sprint, standing broad jump, 1RM seated single leg press, both
		13		20.8					Knee depth jump		
		13		20.8					Ankle depth jump		
Asadi (2012) [66]	Yes	8	Male	20.2	2	6	45	1200	DJ	No	- COD Illinois Agility and T-test
		8		20.3			NA		CMJ		
Berger (1963) [67]	No	20	Male	NR	3	7	NA	210	SJ (50–60% of 10RM)	No	- CMJ jump height
		19							CMJ		
Blakey and Southard (1987) [68]	Yes	11	Male	18–21	2	8	110	500	DJ	RT	- 1RM leg press - Margaria test power level
		10					40		DJ		
		10					NA		Vertical jumps		
Bogdanis et al. (2019) [69]	Yes	7	Mix	18.2–25.8	2	6	NR	1800 FCPL	Mix bilateral	RT	- CMJ, unilateral, and index, DJ (30 cm) height and contact time, RSI, max isometric force (N), RFD (N.s), 1RM leg ext and curl (Sum of right and left, max force bil index)
		8						900 FCPL	Mix unilateral		
Bouguezzi et al. (2020) [27]	Yes	7	Male	11.2	2	8	NA	1360	Mix SSC	No	- Sprint 5 and 20 m, COD Illinois Agility, CMJ, RSI, kicking distance
		8		11.3					Mix non-SSC		
Byrne et al. (2010) [70]	Yes	6	Male	23.8	2	8	40 (height)	660	DJ (counter)	No	- CMJ (cm), RSI 20, 30, 40, 50, and 60, and trained height, inter-individual responses
		6		20.8			30 (RSI)		DJ (bounce)		
Chottidao et al. (2022) [71]	Yes	12	Male	15.5	3	8	20	2220	Mix	No	- Vertical leg stiffness, jump power, RFD (peak and average), jab and cross punch (velocity and force), reaction and movement time
		12		15.6			NA		Mimic rope jump		
Clutch et al. (1983) [72]	Yes	12	Male	20.9	2	4	NA	320	CMJ	RT	- Leg strength (1RM squat), vertical jump (cm), max iso knee extension (125°) (N)
							30		DJ		
							75–110		DJ		
Cronin et al. (2003) [73]	Yes	14	Mix	23.1	2	10	NA	804	Bungy squat jump	No	- Strength and power (EMG, kg, mean and peak velocity) (N and W), single leg jump (cm), lunge test (s), COD T-test (s)
		14							Non-bungy squat jump		
Dello Iacono et al. (2017) [74]	Yes	9	Male	23.4	2	10	25	1028	DJ unilateral/vertical	No	- CMJ (cm, GRF, vertical impulse, leg-spring stiffness, contact time, RSI, and total time), COD (total time, 10 m time, time to perform a turn, step length (0–1.2–4), step frequency, contact time first in 10 m and to turn)
		9							DJ unilateral/horizontal.		
Earp et al. (2015) [75]	Yes	9	Male	18–35	3	8	NA	872	Jump squat parallel	No	- Distal, mid, proximal portion and changes in Sum of quadriceps, vastus lateralis, intermedius, medialis, and rectus femoris, 1RM to BM ratio
		9							Jump squat volitional		
Emamian et al. (2022) [76]	Yes	15	Male	27.6	3	6	60	756	CMJ + box jumps	NR	- CMJ height, shoulder angular velocity, hip angle, hip angular velocity, knee angle, and knee angular velocity at take-off, hip, knee, and ankle angle at the end of the eccentric phase
		15		26.2					CMJ + box jumps (no arms)		
		15		27.1					CMJ + box jumps (no knee)		
Gehri et al. (1998) [77]	Yes	11	Mix	20	2	12	40	704	DJ	No	- SJ, CMJ, and DJ (height and positive energy)
		7		19.5			NA		CMJ		
Gonzalo-Skok et al. (2019) [78]	Yes	9	Male	13.3	2	6	20	960	Mix—vertical/bilateral	No	- 5, 10, 25 m sprint, CMJ, CMJ (left and right), horizontal jump (left and right), COD (V-cut and COD 180), dorsiflexion (right and left), SEBT (left and right, anterior and posterior)
		9		13.2			10		Mix—horizontal/unilateral		
Hawkins (1978) [79]	Yes	10	Male	NR	2/3	6	40–90	552	DJ—optimal height	No	- Sargent jump, standard depth jump, knee extensor, plantar flexor (1RM), inter-individual data reported
		10					40–90		DJ—less height and loaded		
		8					40–90		DJ—less height		
Hoffman et al. (2005) [80]	Yes	15	Male	19.8	2	5	NA	160	SJ—load	No	- BM, 1RM squat and power clean, 40-yard sprint, COD T-test, vertical jump, SJ 70% RM (power and EMG)
		16							SJ—concentric load		
Holcomb et al. (1996) [81]	Yes	10	Male	NR—college age	3	8	NA	1728	CMJ	No	- CMJ (height and peak power) - SJ (height and peak power)
		10					40, 50, 60		DJ (ankle, knee, and hip)		
		10					40, 50, 60		DJ		
Hori et al. (2008) [82]	No	10	Male	23.7	2	8	NA	576	Non-braking weighted SJ	No	- CMJ, SJ, and weight SJ (W and W.kg), RSI, jump and reach, con and ecc squat (1RM), isometric quadriceps and hamstrings strength (10°, 30°, 50°, 70°, 90°), isokinetic con/ecc quads and hams strength (60°/s, 180°/s, 360°/s)
		10		24.8					Braking weighted SJ		
Hortobagyi et al. (1990) [83]	Yes	15	Male	13.4	2	10	NA	2600	Mix—Vertical	No	- BM, height, thigh and calf girth, SBJ, five bound test, vertical jump, vertical jump one leg after a 3-step run-up and back throw over head of a 4 kg ball (cm)
		15							Mix—Horizontal		
Khoadei et al. (2017) [84]	Yes	7	Male	20.1	3	4	NA	1480	Mix-Assisted elastics	No	- Sprint time (10, 10–20, 20–30, and 30 m) - Agility T-test and Illinois (s)
		9		20.9					Mix—Resisted elastics.		
		8		20.9					Mix		
King and Cipriani (2010) [85]	Yes	11	Male	15.3	2	6	NA	1296	Mix—Sagittal plane	No	- Vertical jump (cm)
		10		15.1					Mix—Frontal plane		
Kusuma et al. (2020) [86]	Yes	11	Male	15–17	3	8	NA	NR	Rope jump	NR	- Inter-individual responses; responders, VO_2_ max (mL/kg/min), anaerobic threshold (bpm)
		11							High jump		
Laurent et al. (2020) [87]	No	11	Mix	19–26	2	10	30–40	2980	Mix—Bounce DJ	No	- DJ 20, 40, 60 cm (height, contact, RSI), MVC torque, CMJ (height), tendon stiffness index, Achilles tendon CSA
		11							Mix—Counter DJ		
Loturco et al. (2020) [88]	Yes	13	Male	18.5	3	2	NA	180	SJ—traditional weight	No	- SJ and CMJ (cm), SJ, half squat power (W.kg), 5, 10, and 20 m sprint, COD zig-zag
		12							SJ—elastic band		
Loturco et al. (2015) [8]	Yes	12	Male	18.2	2, 4, 5	3	NA	512	CMJ—vertical	No	- Vert and horizontal jump (height, peak force), 10 and 20 m sprint and accel, effect to speed and accel
		12		18.5					SLJ—horizontal		
Machado et al. (2019) [89]	Yes	8	Male	38	2	8	45	2880 s	SJ	No	- 5 km trial sport-specific
		8		39			45		DJ		
Makaruk et al. (2014) [90]	Yes	12	Male	22.2	3	6	20–30–40–60–76–84–91	3888	Mix—Acyclical	No	- CMJ, repeated CMJ, DJ (force, height, knee flexion degree, landing time)
		12		22.7					Mix—Cyclical		
Makaruk et al. (2011) [91]	Yes	16	Female	20.6	2	12	15–20	6424 FCPL	Mix—Unilateral	No	- Wingate peak power (W), five alternate leg bounds (m), CMJ, and UCMJ (W and m)
		18		20.9			30–35		Mix—Bilateral		
Manouras et al. (2016) [92]	Yes	10	Male	20.7	1	8	40	680	Mix—Vertical	No	- 10 and 30 m sprint, COD right and left side, horizontal and vertical jump
		10		19.1					Mix—Horizontal		
Markovic et al. (2013) [93]	Yes	12	Male	23.7	3	8	NA	1404	CMJ—Unloaded	No	- 1RM squat, SJ, and CMJ (GRF, concentric time, height, power maximal, mean, etc.)
		12							CMJ—Negative elastic		
		12							CMJ—Positive elastic		
		11							CMJ—Vest, change inertia		
Markovic et al. (2011) [94]	Yes	10	Male	11	3	7	NA	1260	CMJ—Deloaded machine	No	- Quadriceps peak torque, CMJ from −30% to +30% BW (Mechanical behavior parameters)
		10							CMJ—Loaded dumbbells		
Marshall and Moran (2013) [95]	Yes	34	Male	22	3	8	30	768	DJ—Bounce	No	- CMJ
		35							DJ—Countermovement		
Mastalerz et al. (2009) [96]	Yes	12	Male	22–24	5	4	NR	800	Mix—Inclined plane	No	- Knee ext power (30, 60, 180, 240°/s), CMJ (power), EMG concentric vastus lateralis and rectus femoris (average and mean)
		12							Mix—Vertical		
Masterson and Brown (1993) [97]	Yes	10	Mix	20.2	3	10	NA	1620 s	Rope jump	No	- CMJ (W), 50-yard sprint, Wingate (mean and maximal), 1RM (leg press bench press)
		12		20.3				660 reps	CMJ		
Matavulj et al. (2001) [98]	Yes	11	Male	15–16	3	6	50	540	DJ—100 cm	No	- CMJ, RFD (knee and hip extensors), MIF (knee and hip extensors)
		11					100		DJ—50 cm		
Mazurek et al. (2018) [99]	Yes	14	Male	20	2–3	5	20, 40, 60, 76	1218	Mix—RSI fast SSC	Yes—RT	- Anthropometric, aerobic capacity, RSI, SJ, CMJ, and SJ (height)
		12							Mix—height, low SSC		
McBride et al. (2002) [100]	No—1RM squat ratio	9	Male	24.2	2	6	NA	Ind	SJ—80%1RM	No	- Anthropometric, 1RM squat, 30 SJ 30%, 55%, 80% (height, peak force, power peak, EMG vastus lateralis), COD agility, 5, 10, and 20 m sprint
		10		21.6					SJ—30%1RM		
McClenton et al. (2008) [101]	Yes	10	Mix	22.1	2	6	NA	139	Mix- Vertimax machine	No	- Vertical jump (cm)
		10		21.3			50–100	137	DJ		
McCormick et al. (2016) [102]	Yes	7	Female	16.3	2	6	NA	1296	Mix—Frontal plane	No	- CMJ, SLJ, right and left lateral hop - COD right and left lateral shuffle test
		7		15.7					Mix—Sagittal plane		
McCurdy et al. (2005) [103]	Yes	NR	Male	20.7	2	6	NA	>360 NCR	Mix—Unilateral	Yes—RT	- Bilateral and unilateral (squat, vertical jump height, absolute and relative power) - Margaria–Kalamen stair climb
		NR	Male						Mix—Bilateral		
		NR	Female						Mix—Unilateral		
		NR	Female						Mix—Bilateral		
		NR	Mix						Mix—Unilateral		
		NR	Mix						Mix—Bilateral		
McGuigan et al. (2003) [104]	Yes	9	Male	24.2	2	8	NA	Ind	SJ—30%1RM	No	- Weight, BM, 1RM squat, myosin heavy chain and fibers (Type I, IIa, and IIb), percentage, mobility, and change of titin-1 and titin-2
		9		21.2					SJ—80%1RM		
Mirzaei et al. (2014) [105]	Yes	10	Male	20.7	2	6	45	1200	DJ	No	- Muscle soreness, 24, 48 h post (rectus, biceps femoris, and gastrocnemius), vertical jump and SLJ, 20 and 40 m sprint, COD T-test and Illinois, 1RM leg press
		10		21.2			NA		CMJ		
Mirzaei et al. (2013) [106]	Yes	9	Male	20.5	2	6	45	1200	DJ	No	- Isometric knee ext EMG (vastus medialis and rectus femoralis), CMJ (height)
		9		20.6			NA		CMJ		
Ramirez-Campillo et al. (2018) [107]	Yes	25	Male	13.9	2	7	30	906	DJ—30 cm	No	- CMJ (height), 20 cm RSI, 5 multiple bounds, 20 m sprint, COD, 5RM squat, 2.4 km trial time, kicking distance
		24		13.1			Optimal RSI		DJ—Optimal (10 to 40)		
Ramirez-Campillo et al. (2015) [26]	Yes	10	Male	11.6	2	6	NA	1610	Mix—Vertical	No	- CMJ vert and horizontal, 20 cm DJ (RSI), multiple 5 bound, kicking velocity, 15 and 30 m sprint-test, and Yo-Yo recovery and stance eyes open and closed (medial lateral and anterior posterior)
		10		11.4				1610	Mix—Horizontal		
		10		11.2				1440	Mix—Vertical/Horizontal		
Ramirez-Campillo et al. (2015) [108]	Yes	12	Male	11	2	6	NA	2160 FCPL	Mix—Bilateral	No	- CMJ horizontal and vert (right, left, and bilateral), 20 cm DJ (RSI), multiple 5 bound, kicking velocity, 15 and 30 m sprint, T-test, and Yo-Yo recovery and stance eyes open and closed perturbed (medial lateral and anterior posterior)
		16		11.6				1080 FCPL	Mix—Unilateral		
		12		11.6				1440 FCPL	Mix—Bilateral/Unilateral		
Rosas et al. (2016) [109]	Yes	21	Male	12.3	2	6	NA	1152	Mix	No	- CMJ vert and horizontal (right, left, and bilateral), 20 cm DJ (RSI), kicking velocity
		21		12.1					Mix—handheld haltered		
Ruffieux et al. (2020) [110]	Yes	13	Female	20.4	2	6	37	720	CMJ (80%) + DJ (20%)	Yes—regular	- CMJ, CMJ arm swing, CMJ run-up and DJ
		13		22					DJ (80%) + CMJ (20%)		
Sheppard et al. (2008) [111]	Yes	8	Mix	21.8	3	5	NA	705	CMJ—load eccentric	Yes—volleyball	- CMJ (height, peak velocity, peak force, and power)
		8							CMJ—without load		
Singh et al. (2018) [112]	Yes	8	Mix	23	2	6	30–40	240	DJ—low to high	Yes—RT	- 10 and 20 m sprint - 505 COD
		8					70–85		DJ—high to low		
Singh and Singh (2013) [113]	Yes	20	Male	18–21	2	10	20, 25, 30, 35, 40	1200	DJ—Vertical	NR	- 45.72 m sprint
		20							DJ—Horizontal		
		20							DJ—Vertical/Horizontal		
Singh and Singh (2012) [114]	Yes	20	Male	19.9	2	10	Optimal 20–40	1200	DJ—Vertical	NR	- DJ (height)
		20							DJ—Horizontal		
		20							DJ—Vertical/Horizontal		
Singh and Singh (2012) [115]	Yes	20	Male	19.9	2	10	Optimal 20–40	1200	DJ—Vertical	NR	- Running long jump
		20							DJ—Horizontal		
		20							DJ—Vertical/Horizontal		
Sotiropoulos et al. (2022) [116]	Yes	11	Female	23.8	1–2	8	Optimal RSI	600	DJ—Optimal RSI	Yes—RT	- 1RM half squat, SJ, CMJ, and CMJ no arms (height), DJ 20, 30, 40, 50, 60, and 70 cm (height, contact time, and RSI)
		11					25% high		DJ—25% high		
		11					25% low		DJ—25% less		
Staniszewski et al. (2021) [117]	Yes	13	Male	21	5	4	14–28	1600	Box upward + vertical jumps	Yes—PE classes	- Muscle torque (hip and knee flexors, extensors, and plantar flexor), CMJ (force, velocity, power, height, and range of swing), creatinkinase
		13							Box downward + vertical jumps		
Stern et al. (2020) [118]	Yes	11	Male	17.6	2	6	30–40	576	Mix—Unilateral	RT split	- 1RM squat and split, CMJ, single leg CMJ, SLJ, RSI (left, right, and reactive), 10 and 30 m sprint, COD 505 left and right (time, deficit)
		12					15–20		Mix—Bilateral	RT squat	
Stien et al. (2020) [119]	Yes	18	Female	21.3	2–3	8	NA	1380	Mix—elastic band assisted	No	- Squat 40, 60, 80% (con velocity), SJ (height), muscle thickness vastus lateralis
		18		20.9					Mix—elastic band resisted		
Strate et al. (2022) [120]	Yes	16	Female	21.3	2–5	8	NA	1380	Mix—elastic band assisted	No	- Training attendance, CMJ (height), 40 m sprint (speed, accel, and time)
		17		20.9					Mix—elastic band resisted		
Taube et al. (2012) [121]	Yes	11	Mix	24	3	4	30, 50, 75	396	DJ—Bounded	No	- DJ 30, 50, and 75 cm (GCT, height, RSI, H-reflex/M-wave ratio, M-wave, soleus, rectus femoris, gastrocnemius; tibialis anterior muscle activity and hip, knee, and ankle flexion angle)
		11		25			30		DJ—Counter		
Thomas et al. (2009) [122]	Yes	6	Male	17.3	2	6	40	580	DJ	No	- Vertical jump, 505 COD, 5, 10, 15, and 20 m sprint
		6					NA		CMJ		
Trzaskoma et al. (2010) [123]	Yes	10	Male	22.1	4	3	NA	1176	Pendulum “natural” take off	No	- CMJ (height, power, and Sum of torques hip and knee extensors), 1RM squat
		10		22.6			NA		Pendulum “impact” take off		
Watkins et al. (2021) [124]	Yes	8	Male	18.9	2	3	30	300	Mix—Horizontal	No	- 10, 20, 30 m sprint (time) - VO, Vmax, F0 (N), F0rel (N·kg), Pmax (W), Prel (W·kg), Sfv, Srel, RFmax (%), DRF (%)
		8					20–60		Mix—Vertical		
		12		19.8			30		Mix—Horizontal		
		12					20–60		Mix—Vertical		
Weakley et al. (2021) [125]	Yes	16	Male	20.8	3	4	NA	108 SJ + 72 horizontal	SJ barbell + horizontal	Yes—RT + others	- Jump (height, peak velocity, peak power, mean power, mean force, and impulse)
		13		21.4					SJ hexagonal + horizontal		
Weltin et al. (2017) [126]	Yes	12	Female	21	3	4	NA	2890 FCPL	Unilateral lateral jumps	No	- Lateral unilateral jumps (reduced/increased degrees of trunk internal knee internal rotation, reduced trunk degrees, step width)
		12		22			45	>3940 FCPL	Mix—Bilateral vertical		
Wilson et al. (1993) [127]	Yes	13	NR	22.1	2	10	20–80	>540	DJ	No	- CMJ, SJ, leg ext strength isokinetic, 30 m sprint and 6 s cycling, isometric force, and RFD
		13		23.7			NA		SJ—Loaded		
Yang et al. (2020) [128]	Yes	20	Mix	13.4	3	12	NA	88,560–95,040	Rope jump—freestyle	No	- SLJ, 1RM left right-hand grip, flexibility, waist circumference, BMD
		20		13.5					Rope jump—traditional		
Young et al. (1999) [129]	Yes	11	Male	19–34	3	6	Max height	468	DJ—for height	No	- Standing vertical jump, run-up jump, SJ (height, dynamic and isometric strength relative to BM), DJ (height and RSI)
		5					Max RSI		DJ—for RSI		

Note: abbreviations are ordered alphabetically. BM: body mass; BMD: bone mass density; CMJ: countermovement jump; COD: change of direction; DJ: drop jump; Dur: duration of plyometric jump training (weeks); EMG: electromyography; FCPL: foot contacts per leg; Freq: frequency of plyometric jump training (sessions per week); GCT: ground contact time; MIF: maximal isometric force; NA: not applicable; NR: not reported; PE: physical education; RFD: rate force development; RM: repetition-maximum; RSI: reactive strength index; RT: resistance training; SEBT: star excursion balance test; SJ: squat jump; SLJ: standing long jump; SSC: stretch-shortening cycle.

**Table 2 sports-11-00150-t002:** Participant characteristics included in eligible articles.

*Sex*	Male	71.0%	*Age*	≥18 years old	72.5%	*Physical performance level*	High	14.5%	*Sport practiced*	Team sports	34.8%	*PJT previous experience*	Experience	20.3%	*Training period*	In-season	17.4%
	Female	10.0%		<18 years old	24.6%		Moderate/normal	71.0%		Individual sports	7.3%		No experience	43.5%		Pre-season	13.0%
	Mix	17.4%		NCR	2.9%		Low	5.8%		Mixed	10.1%		Mixed	1.4%		Off-season	2.9%
	NCR	1.6%					Mix	1.4%		Non-Competitive	33.3%		NCR	34.8%		Non-Competitive	56.5%
							NCR	7.3%		NCR	14.5%					NCR	24.6%

PJT: plyometric jump training; NCR: not clearly reported among eligible articles.

**Table 3 sports-11-00150-t003:** Plyometric jump training prescription characteristics.

*Surface*	Soft	17.3%	*Dose*	Reported	97.1%	*Habitual training*	Added	34.8%	*Combined?*	Yes	17.4%	*Duration*	≥6 weeks	79.7%	*Frequency*	1 day/week	1.4%
	Unstable	1.4%		No reported	2.9%		Replaced	11.6%		No	75.4%		<6 weeks	20.3%		2 days/week	55.1%
	Machines	2.9%					No previous training	17.4%		NCR	7.2%					3 days/week	30.4%
	Mat/parquet	1.4%					NCR	36.2%								≥4 days/week	4.4%
	NCR	73.9%														Mixed	8.7%
*Intensity*	Maximal	60.9%	*Progressive overload*	Volume	29.0%	*Tapering*	No	7.2%	*Rest/Sets*	> 120 s	31.9%	*Rest/Sessions*	≥48 h	33.3%			
	Submaximal	13.0%		Intensity	10.1%		Yes	11.6%		≤ 120 s	46.4%		NCR	50.7%			
	NCR	26.1%		Technique	10.1%		NCR	81.2%		NCR	21.7%						
				Mixed	20.3%												
				No overload	21.7%												
				Yes, no report	5.8%												
				NCR	2.9%												

Surface: type of surface on which training intervention were performed; dose: studies that reported total dose used in their training intervention (could be reported as foot contacts per leg, number of jumps, time, velocity, strength, etc.); habitual training: studies that reported if intervention period was added or replaced by their usual training; combined?: studies in which PJT was combined with another type of training; duration: intervention duration; frequency: total number of training used per week during training interventions; intensity: PJT training intensity reported; progressive overload: overload followed during PJT intervention period; tapering: reduction of any training variables previous post-tests; rest/sets: rest between sets during PJT exercises; rest/sessions: rest between PJT training sessions; NCR: not clearly reported among eligible articles.

**Table 4 sports-11-00150-t004:** Evidence-gap map of HPC adaptations related to the type of PJT exercise.

		OUTCOMES												
		Strength	Vertical Jump	Horizontal Jump	Sprint	COD/Agility	Power	Asymmetry	SSP	Physiological Changes	Biomechanical Changes	Flexibility	Balance	Aerobic Capacity
COMPARATORS	Hip DJ vs. Knee DJ ^a^	1	1	1	1									
	Hip DJ vs. Ankle DJ ^a^	1	1	1	1									
	Knee DJ vs. Ankle DJ ^a^	1	1	1	1									
	DJ vs. CMJ	3	6	1	2	3	3			1				
	Loaded vs. Unloaded	2	3				1							
	Bounce DJ vs. Counter DJ ^b^	7	9		2	2	1		1	2	1			1
	Bilateral vs. Unilateral	3	5	2	2	2	3	1			1		1	
	Fast SSC vs. Slow SSC ^c^	1	3		1	1	1	1		2				1
	Cyclical vs. Acyclical	3	2		1		2		1	2	1			
	Eccentric overload vs. Plyometric	1	1			1	1							
	Vertical vs. Horizontal	1	4	3	4	3	1		2	2			1	
	CMJ vs. No arms CMJ		1								1			
	Vert + Bil vs. Hor + Uni		1		1	1						1	1	
	Loaded vs. concentric load	1	1		1	1	1			1				
	Eccentric braking vs. No braking	2	2				2							
	Assisted vs. Resisted	3	3		2	1	2			3	1			
	Sagittal vs. Frontal		2	1		1								
	Bands vs. Traditional weight		1		1	1								
	SJ vs. DJ	1	1		1				1					
	Inclined vs. Vertical	1	1				1							
	SJ 80% vs. SJ 30%	2	1		1	1	1			1	1			
	Machine vs. DJ		1											
	Box jump upward vs. downward	1	1				1			1	1			
	Handheld altered vs. plyometric	1	1	1					1					
	Traditional barbell vs. hexagonal	1	1				1							
	Rope jump traditional vs. freestyle	1		1						1		1		

Yellow: one article; Orange: two-three articles; Green: four or more articles; White: no article available; ^a^ Participants were asked to focus on specific joints; ^b^ Denotes that studies either compared different DJ heights (e.g., individualized vs fixed), or different DJ technique (e.g., bounce jump [i.e., focus on reduced foot-ground time contact] vs counter jump [focus on jump height]); ^c^ Denotes that studies compared fast vs slow SSC jump exercises using different approaches (e.g., participants were asked to focus on reduce joint range of movement vs increase joint range of movement; participants were asked to focus on reduce foot-ground time contact vs focus on increase jump height). CMJ: countermovement jump; DJ: drop jump; HPC: human physical capabilities; PJT: plyometric jump training; SJ: squat jump; SSC: stretch-shortening cycle; SSP: sport-specific performance.

## Data Availability

The article contains all of the data produced or analyzed during this investigation as table(s), figure(s), and/or electronic supplemental material(s). Any further data requirements may be directed to the authors upon a reasonable request.

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
