# Peer review of "Plyometric Jump Training Exercise Optimization for Maximizing Human Performance: A Systematic Scoping Review and Identification of Gaps in the Existing Literature"

_sports, 2023, doi:10.3390/sports11080150_

Round 1

Reviewer 1 Report

Thank you for inviting me to review the work by  Dudagoitia and colleagues on the existing gaps in the scientific literature regarding plyometric jump training for enhancing physical performance. The study is methodologically remarkable, well-documented, and follows a suitable structure, which facilitates a concise and comprehensive reading. Despite the presence of several recent reviews on the effects of plyometric training on human performance, the authors effectively justify the need to identify gaps in the literature and emphasize the necessity of periodic updates to address the exponential increase in publications in this field of knowledge.

Here are some suggestions to strengthen the study presented by the authors:

  1. The title does not reflect the main objective of the study. Please consider adding "identification of gaps in the existing literature" to avoid misleading the reader.
  2. The authors acknowledge the omission of a quality analysis of the studies but fail to adequately justify this decision. I suggest considering the inclusion of such an analysis (e.g., PEDro, Downs and Black scales) to further strengthen their work. In any case, citing a previous review study on anterior cruciate ligament is not a sufficient argument.
  3. In the conclusions, the existing gap in the literature regarding the effects of plyometric training on aerobic capacity, flexibility, balance, and asymmetries is highlighted. However, the rationale for how this type of training could improve these qualities is not clear to me, nor is it justified in the introduction. Additionally, the references cited in the introductory section are not original research studies and do not address this question.
  4. Consistent with the previous comment, it is suggested that the authors remove citations to previous reviews when justifying an argument and instead cite the original article(s).

Minor comments:

  1. Page 25 line 33: please add the "h" in the word "both"

Reviewer 2 Report

Dear authors I understand a lot of work have been done on this manuscript but I see some problems:

1)     There are many studies about PJT and HPC that are not included, I have doubts that the seach strategy is correct.

2)     The document is not well presented, table 4, figure 1, sometimes spaces between parragrafs other times no, the same with italics…

3)     I do not see what this manuscript provide to the scientific comunity. The title of the manuscrip is “Plyometric Jump Training Exercise Optimization for Maximizing Human Performance: A Systematic Scoping Review” however results and conclusion sections, in the abstract, do not offer to the reader any information about how much can human performance be improved by any kind of PJT. Sentences like: “In addition, individualized and mixed different kinds of jumps with different characteristics may suppose better strategies for strength and professional coaches to improve strength, vertical jump and sprint performance.” Are not supported by the results presented in the abstract.  “Authors extracted these conclusions from PJT type exercises more studied in the literature, with four or more studies conducted” Authors should present the results and then conclusions, otherwise those sentences are subjective. At the end of the introduction authors said: “This review approach would add valuable information to the literature for practitioners and applied researchers” the variety in individual and mixed different kinds of jumps is huge, how can practiciones, coaches or applied researches implemented this suppose valuable information if it is not specific?

Minors:

To indicate de meaning of all abreviations (DJ, SJ…..) in the abstract, manuscript, figures, tables. And use them, examples page 7 PJT instead of pliometric jump training.

Instead to give some examples of seach: “Examples of combinations included: “ballistic” AND “training”; (“ballistic” OR “plyometric” OR “explosive”)” please to add all the combinations used for search strategy.

Figure 1 can be done better. Beside the total records identified (12,503) is different that 10,546 (indicated in the results section).

Table 1, 2, 3 , 4 must be improved.

Best Regards

Reviewer 3 Report

Initially, I would like to congratulate the authors for developing this study.

- Graphs and charts will need to be fine-tuned. The content is not that good.

-It would be crucial to indicate the practical applications of this article.

-Also, what volume, intensity, and types of jumps can be applied at different competitive levels and age groups? The authors could explain about the volume, intensity, type of jump from beginner and intermediate athletes and even high-performance athletes, considering the positions, modalities, and profile of each athlete.

Round 2

Reviewer 2 Report

Dear Authors, 

The article is better now

Best Regards